# Prevalence of Non-Affective Psychoses in Individuals with Autism Spectrum Disorders: A Systematic Review

**DOI:** 10.3390/jcm8091304

**Published:** 2019-08-24

**Authors:** Riccardo De Giorgi, Franco De Crescenzo, Gian Loreto D’Alò, Nicola Rizzo Pesci, Valeria Di Franco, Corrado Sandini, Marco Armando

**Affiliations:** 1Department of Psychiatry, Medical Science Division, University of Oxford, Oxford OX3 7JX, UK; 2Oxford Health NHS Foundation Trust, Oxford OX3 7JX, UK; 3Pediatric University Hospital-Department (DPUO), Bambino Gesù Children’s Hospital, 00165 Rome, Italy; 4Department of Epidemiology, Lazio Regional Health Service, 00147 Rome, Italy; 5School of Hygiene and Preventive Medicine, University of Rome Tor Vergata, 00133 Rome, Italy; 6Department of Clinical Neurosciences, University Vita-Salute San Raffaele, 20132 Milan, Italy; 7Department of Anesthesiology and Intensive Care Medicine, Catholic University of The Sacred Heart, 00168 Rome, Italy; 8Developmental Imaging and Psychopathology Laboratory, Department of Psychiatry, University of Geneva School of Medicine, CH-1211 Geneva, Switzerland

**Keywords:** Pervasive developmental disorders, Autism spectrum disorders, Schizophrenia Spectrum Disorders, Non-affective psychoses, Systematic review

## Abstract

Autism spectrum disorders (ASD) and non-affective psychoses such as schizophrenia are commonly acknowledged as discrete entities. Previous research has revealed evidence of high comorbidity between these conditions, but their differential diagnosis proves difficult in routine clinical practice due to the similarities between core symptoms of each disorder. The prevalence of comorbid non-affective psychoses in individuals with ASD is uncertain, with studies reporting rates ranging from 0% to 61.5%. We therefore performed a systematic review and pooled analysis of the available studies reporting the prevalence of non-affective psychosis in ASD. Fourteen studies, including a total of 1708 participants, were included, with a weighted pooled prevalence assessed at 9.5% (95% CI 2.6 to 16.0). In view of significant heterogeneity amongst the studies, subgroup analyses were conducted. We observed higher prevalence of non-affective psychoses among ASD inpatients versus outpatients, when operationalised criteria were used, and in studies with smaller sample sizes, whereas the figures were comparable between children and adults with ASD. Our results suggest that future studies involving larger samples should implement both operationalized criteria and specific scales for the assessment of psychotic symptoms in individuals with ASD. A deeper understanding of both differential and comorbid features of ASD and non-affective psychosis will be required for the development of optimized clinical management protocols.

## 1. Introduction

Historically, non-affective psychoses (i.e., schizophrenia spectrum disorders) and autism spectrum disorder (ASD) have not always been conceptualised as separate illnesses. Bleuler categorised autism as a form of schizophrenia and theorized autistic symptoms as “infantile wishes to avoid unsatisfying realities and replace them with fantasies and hallucinations” [1]. Others viewed autism as a form of childhood-onset schizophrenia [2]. In 1943, Kanner described a case series of children presenting with autistic disturbances of affective contact and named this condition as early infantile autism [3]. The terms “autism” and “schizophrenia” were used interchangeably until the 1970s, when Rutter [4] and Kolvin [5] proposed a distinction between the two disorders.

This dichotomy has evolved over the last decades and has been implemented in current diagnostic manuals. In the DSM-5, non-affective psychoses are reported under the “schizophrenia spectrum disorders” label and include schizophrenia, other psychotic disorders, and schizotypal personality disorder that find a common denominator in the presence of delusions, hallucinations, disorganized thinking, disorganized behavior, and negative symptoms [6]. ASD on the other hand are defined as early-onset persistent condition characterised by a combination of deficient social skills, restricted and repetitive behavior, and difficulties in verbal and non-verbal communication [6]. There are several strong arguments in favour of a nosological distinction between the two disorders. Notably, ASD symptoms must be present in the early developmental period and the disorder can be diagnosed as early as 3 years old [6], whereas non-affective psychoses typically occur between late adolescence and early adulthood and are rare during childhood [7].

However, there is also evidence of several shared features between the two conditions. From a neurobiological perspective, rates of heritability are both estimated to be high at around 50–80% [8,9,10]. Interestingly, together with high levels of heritability within each disorder, there is evidence of relatively high levels of heritability between the disorders [11,12,13,14]. Recent genetic studies have demonstrated that common copy number variants (CNVs; variations of DNA sequence in the genome) can contribute to the risk for both schizophrenia and ASD [15,16,17,18]. Also, routine clinical practice suggests that ASD and psychoses may co-occur at a higher frequency than what would be expected by chance.

The prevalence of non-affective psychosis in ASD is unclear and evidences are mixed. Previous reports described a rate of non-affective psychosis in ASD ranging from 0% to 34.4% [19], while other studies reported a prevalence as high as 61.5% [20]. Conversely, a recent meta-analysis showed that individuals with non-affective psychoses have higher levels of autistic symptoms compared to healthy controls (SMD = 1.39, 95% confidence interval (CI) = 1.11 to 1.68) and lower levels of autistic symptoms compared to individuals with autism (SMD: −1.27, 95%CI = −1.77 to −0.76) [21].

This gap of knowledge has two significant clinical implications. Firstly, it can lead to a potential under-estimation of the prevalence and impact of psychotic symptoms in individuals with ASD. Consequently, this results into a lack of clear clinical guidelines for the management of ASD individuals experiencing psychotic symptoms, with a potential for under-treatment. Therefore, the aim of this study is to systematically review and synthesise the current evidence on the prevalence of non-affective psychosis in individuals with ASD.

## 2. Methods

We performed a comprehensive literature search (Appendix A) of the PubMed/MEDLINE, Web of Science, CINHAL, databases from the date of their inceptions until May 2019. We included studies that evaluated the prevalence of non-affective psychoses in individuals with ASD with no age limit, but we excluded studies on mood disorders with psychotic features such as major depression with psychotic symptoms and bipolar disorder with psychotic symptoms.

Four researchers (C.S., G.L.D., N.R.P., V.D.F.) independently screened titles and abstracts from all databases for relevance. Two researchers (G.L.D., N.R.P.) retrieved and assessed the full-texts to determine eligibility. Disagreements were resolved by consensus with another researcher (F.D.C.). For the included study, three researchers (F.D.C., G.L.D., N.R.P.) independently extracted data and documented information about the publication (i.e., authors’ names, year of publication, country) and patients’ characteristics (i.e., gender, age, diagnostic criteria, symptoms severity, setting, outcomes). The quality and potential sources of bias for each study were evaluated with the Newcastle Ottawa Scale (NOS) (Appendix A).

Finally, we discussed the included studies separately and quantitatively analysed the prevalence of all non-affective psychoses (including schizophrenia, delusional disorder, other schizophrenia spectrum disorder or psychotic disorder not otherwise specified) in patients with ASD. We calculated the weighted pooled prevalence mean for all studies using STATA 15 and we performed subgroup analyses by computing the weighted pooled prevalence mean for the categories “setting” (inpatients and outpatients), “age” (<18 years old and ≥18 years old), and “diagnostic system” (DSM-III-R or more recent with ICD-10 and ICD-9). Also, as suggested by previous authors [19], because ASD and non-affective psychoses have a prevalence of ~1% of the general population [22,23,24,25], the significance of smaller studies (i.e., fewer than 100 individuals) may be misleading; therefore, we did another subgroup analysis for “sample size” (*N* < 100 and *N* ≥ 100).

## 3. Results

Our literature search followed the Preferred Reporting Items for Systematic Reviews and Meta-Analyses (PRISMA) guideline (Figure 1) and retrieved 1817 records through electronic databases searching and 8 additional records through manual reference screening. Of the total 1825 screened records, 1786 records were excluded because title and abstract did not meet the inclusion criteria. Thirty-nine full-texts were assessed for eligibility (Appendix A): 14 were excluded for not reporting any outcome of interest, 7 because they included participants who did not meet inclusion criteria, 3 for being reviews, and 1 whose data on the outcome of interest were not extractable (totalling 25 full-text article excluded). Finally, 14 articles that reported the prevalence of non-affective psychoses or a formal diagnosis of schizophrenia in patients with ASD were included in this review. The individual studies’ characteristics are reported in Table 1.

The estimated weighted pooled prevalence was 9.5% (95% CI = 2.6 to 16.0), number of patients with ASD (*N*) = 1708, number of cases of non-affective psychoses in ASD (*n*) = 163. The significance of this result is limited as it combines data from different studies with non-homogenous populations. In order to better describe these data, we divided the included papers into two categories: those showing a low (<20%, 10 studies) and a high (≥20%, 4 studies) prevalence of non-affective psychoses in ASD.

### 3.1. Low Prevalence of Psychotic Symptoms/Schizophrenia in ASD (<20%)

Two studies did not identify any individual with ASD and a comorbid non-affective psychosis. Both include comparable populations in terms of age (respectively 24 years and 34.9 years), gender (22.9% in both studies), and setting (outpatients). However, Eaves et al. [26] included only 48 participants diagnosed with ASD according to DSM-IV with a verbal IQ ranging from <34 to >70, whereas Hutton et al. [27] involved a larger sample of 135 people with an ICD-10 diagnosis of ASD and a non-better specified IQ > 30. Similarly Volkmar et al. [28] reported that in a sample of 163 outpatients diagnosed with ASD according to the DSM-III-R, the prevalence of psychosis was very low (0.6%), which is lower than what expected in the general population.

The largest of all retrieved studies (*N* = 414) included a slightly younger population (mean age = 16.2 years) and found a prevalence of psychosis of 2.9% [29].

A slightly higher prevalence of psychosis was reported in three Swedish studies. Stahlberg et al. [30] identified a diagnosis of schizophrenia or other psychotic disorders (excluding bipolar affective disorder) in 7.8% of 129 outpatients with ASDs. Billstedt et al. [31] found a prevalence of psychosis of 7% and 9% in outpatients diagnosed respectively with autism (*N* = 73) and atypical autism (*N* = 35) according to DSM-III-R criteria. Lastly, 12% of 122 people diagnosed with ASD and normal IQ in a community setting were reported as psychotic in a more recent paper [32].

A single study included 54 outpatients with Asperger syndrome; as expected, these showed higher IQ (mean = 102) and prevalence of females (51.8%) and the prevalence of psychosis was of 3.7% [33]. Another study focussed on a diagnosis of infantile autism (mean age = 5.4 years) according to ICD-9 in 118 children admitted to hospital and reported psychosis in 6.6% of them [34]. Finally, Guinchat et al. [35] found a prevalence of psychosis of 9% in a small sample (*N* = 58) of inpatients with ASD and a high percentage (71%) of severe/profound intellectual disability.

### 3.2. High Prevalence of Psychotic Symptoms/Schizophrenia in ASD (≥20%)

Two studies from 2010 reported a higher prevalence of psychotic symptoms measured with specific scales. A sample (*N* = 62) diagnosed with autism and intellectual disability (IQ < 34 in 30 patients, IQ = 35–70 in 32 patients) used the Psychopathology in Autism Checklist (PAC), which scored positively on the psychotic subscale in 25.1% of community patients [36]. Similarly, a larger study of 217 children (mean age = 9.7 years) with ASD in an outpatient setting found a prevalence of psychotic symptoms of 20% on the Schedule for Affective Disorders and Schizophrenia for School-Age Children (K-SADS-E) [37].

One study only included inpatients with an ICD-10 diagnosis of atypical autism (*N* = 89); 34.8% of them presented with psychosis [38]. A very high percentage (61.5%) of people with ASD were formally diagnosed (DSM-IV-TR) with schizophrenia in a small sample (*N* = 26) of inpatients in Italy; however, it should be noticed that the study setting was in a psychiatric intensive care unit where more acute and disruptive psychopathology—in this case, psychotic symptoms—can be reasonably expected [20].

### 3.3. Subgroup Analyses

The subgroup analysis for “setting” showed a high prevalence of non-affective psychoses in ASD for inpatients (20.6%, 95% CI = 0.0 to 43.0) and a lower prevalence in outpatients (9.1%, 95% CI = 2.2 to 15.9).

The subgroup analysis for “age” showed similar prevalence in children (<18 years old: 8.47%, 95% CI = 0.0 to 17.5) and adults (≥18 years old: 10.1%, 95% CI 0.0 to 21.4).

The subgroup analysis for “diagnostic system” showed comparable results between the two systems using operationalised criteria (DSM-III-R or more recent: 11.7%, 95% CI = 0.0 to 26.4; ICD-10: 12.5%, 95% CI = 1.4 to 23.5), but lower prevalence for ICD-9 (3.72%, 95% CI = 0.39 to 7.0), which does not use operationalised criteria.

The subgroup analysis for “sample size” showed higher prevalence in smaller studies (*N* < 100: 19.6%, 95% CI = 3.4 to 35.9) against larger studies (N ≥ 100: 6.36%, 95% CI = 0.0 to 13.2).

## 4. Discussion

In this study, we systematically reviewed and pooled all the available evidence on the prevalence of non-affective psychoses in ASD, which we estimated at 9.56%—in line, though slightly lower than the 12.8% reported in a previous review [19]. Our result could reflect a more accurate estimate of the true prevalence, given that the present metanalysis includes 5 additional publications [20,27,28,29,35] in addition to the 9 [26,30,31,32,33,34,36,37,38] included in the aforementioned review. Moreover, of the additional 5 articles retrieved, 3 [27,28,29] have large sample sizes with an *N* >100. An earlier Lancet’s Seminar in 2014 reported a prevalence of psychotic disorders in autism ranging from 12% to 17%, which is coherent with our results, although the authors did not further comment on this finding [39]. Considering a prevalence of non-affective psychoses in the general population of ~1% [23,25], evidence suggests that the prevalence of non-affective psychoses may be almost ten-fold or higher in people with ASD. Interestingly, another Lancet Seminar from 2018 does not mention the comorbidity of non-affective psychosis in ASD [40], which would suggest a significant under-estimation of the prevalence and impact of psychotic symptoms in individuals with ASD.

Our results suggest that optimal clinical practice should include a routine and careful assessment of psychotic symptoms, in patients with ASD. The use of gold standard clinical instruments for the assessment of both ASD and psychosis is particularly important given the degree of shared or overlapping symptomatology across the two disorders, that complicates differential diagnosis. For example, language difficulties, poverty of speech, formal thought disorder, overvalued ideas, and deficit in interpersonal relationships are often present in both ASD and psychosis [41,42].

Notably, only 2 studies [36,37] employed clinical scales for evaluating psychotic symptoms (PAC: Psychopathology in Autism Checklist and K-SADS-E: Schedule for Affective Disorders and Schizophrenia for School-Age Children respectively). While neither instrument is specifically designed to gauge psychosis, both studies found a higher prevalence of psychoses (25.1% and 20% respectively) in outpatient’s samples. These results would again suggest a significant under-estimation of the prevalence of psychotic symptoms in individuals with ASD and highlight a clear lack of research employing dedicated instruments for detection of sub-threshold psychotic symptoms in individuals with ASD.

### 4.1. Limitations

Our study has a number of limitations, and the estimated mean prevalence should be interpreted with caution.

Despite our broad database search with relatively loose search terms, only 14 studies could be found and were included in the systematic review. However, though the number of studies was small, the total sample size was reasonably large (*N* = 1708) and results were precise and consistent with previous reports [19,39].

We observed significant heterogeneity among the 14 studies that were included, which is why we narratively discussed each study individually and performed subgroup analyses to achieve more homogeneous samples.

Psychoses were far more common in inpatients (20.6%) compared to community samples (9.1%). This is hardly surprising, given that the presence of psychotic symptoms, that are inherently troubling to both patients and families, significantly increases the likelihood of requiring hospitalization. Moreover it is likely that the more closely-monitored environment of a psychiatric ward increases the likelihood of detecting more subtle psychotic symptoms.

The prevalence of non-affective psychoses was only marginally higher in adulthood (10.1%) compared to childhood (8.4%). This is an interesting finding, since traditionally the onset of psychotic disorders is uncommon in children [9], thus suggesting that a young age should not dismiss the assessment of psychosis in children with ASD.

Another meaningful result was the considerably higher prevalence of non-affective psychoses when the diagnosis is guided by operationalized criteria (11.4% and 12.5% for DSM-III or more recent ICD-10 respectively, against 3.7% for ICD-9). No study employed the most recent DSM-5; due to its broader criteria for a “spectrum” diagnosis for both autistic and schizophrenic disorders, it is possible that the prevalence of these conditions may have been underestimated.

As previously discussed, it has been argued that sample sizes below 100 may not capture the real prevalence on non-affective psychoses in ASD19; we found a prevalence of 19.6% in smaller studies versus 6.3% in those with N ≥ 100, in accordance with the existence of an overrepresentation bias in smaller studies.

### 4.2. Future Perspectives

In view of the significant comorbidity of non-affective psychoses in ASD, studies investigating the aetiology of psychotic symptoms in patients with ASD could provide a stronger evidence-base for guiding clinical assessment and treatment. Future epidemiological studies should aim to assess the overall prevalence of psychoses in ASD on large community-based sample sizes, using diagnostic manuals with operationalised criteria and clinical scales that are specific for psychotic symptoms. This would help disentangle the complex psychopathology behind these disorders, thus leading to an improved differentiation between the two diagnoses as well as a better recognition of their comorbidity.

## 5. Conclusions

We found that about 9.5% of patients with ASD present a comorbid non-affective psychosis. These conditions may offer crucial targets for intervention and should be routinely evaluated in the clinical assessment of individuals diagnosed with ASD.

## Figures and Tables

**Figure 1 jcm-08-01304-f001:**
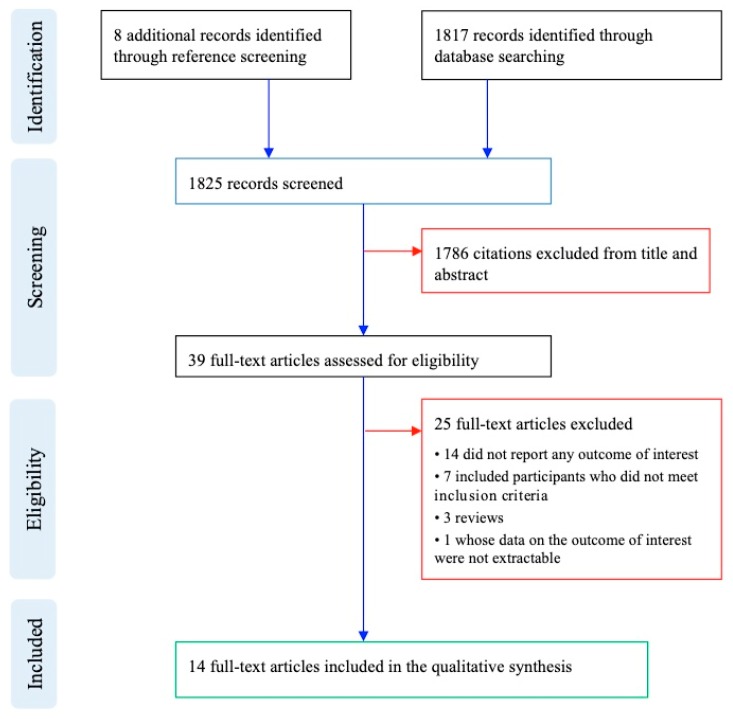
Flow chart.

**Table 1 jcm-08-01304-t001:** Characteristics of the included studies and main findings on non-affective psychosis comorbidity in individuals with ASD.

Study, Year	Country	Diagnosis, *n*	ASD Severity	Age (Mean/Median, SD/Range)	% Female	IQ/Intellectual Disability	Setting	% Psychotic	Scale
Bakken, 2010	Norway	AD and Intellectual Disability (ICD-10), *n* = 62	NR	24.3 (14–57)	27%	IQ < 34, *n* = 30	Specialist outpatient	25.10%	PAC—psychosis subscale *
						IQ 35–70, *n* = 32			
Billstedt, 2005	Sweden	AD (DSM-III-R), *n* = 73	AD: GAF = 22.2 (SD 16.8); Atypical autism: GAF = 18.5 (SD 15.2)	NR	30%	IQ < 50, *n* = 56;	Specialist outpatient	7% (AD); 9% (Atypical autism)	Psychiatrist assessment
		Atypical autism (DSM-III-R), *n* = 35				IQ 50–70, *n* = 42, IQ 71–85, *n* = 17; IQ > 85, *n* = 5			
Eaves, 2008	Canada	ASD (DSM-IV), *n* = 48	CARS = 31.0 (5.9)	24 (19–31)	22.90%	Verbal IQ < 34, *n* = 7; IQ 35–49, *n* = 16; IQ 50–69, *n* = 15; IQ > 70, *n* = 8	Specialist outpatient	0%	NR
Hofvander, 2009	Sweden, France	ASD (DSM-IV), *n* = 122	NR	29 (16–60)	33%	Normal IQ	Specialist outpatient	12%	NR
Joshi, 2010	US	ASD (DSM-III-R), *n* = 217	NR	9.7 (3.6) Range = 3–17	13%	NR	Specialist outpatient	20%	K-SADS-E
Lugnegård, 2011	Sweden	Asperger syndrome, *n* = 54	NR	27 (3.9)	51.85%	Mean IQ = 102 (SD 12)	Specialist outpatient	3.70%	NR
Guinchat, 2015	France	ASD (ICD-10), *n* = 58	GAF = 17.66 (9.05);	15.66 (4.07)	24.13%	Severe/profound ID, *n* = 40 (71.00%)	Specialist inpatient	9%	NR
			CARS = 40.18 (4.76)	Range = 10.9–37					
Abdallah, 2011	Denmark	ASD (ICD-8/ICD 10), *n* = 414	NR	16.28 (4.55)	19.08%	ICD-8/ICD-10 diagnosis *n* = 88 (21.3%)	NR	SCZ (ICD-8/ICD-10) 2.9%	NR
Raja,	Italy	ASD (DSM-IV-TR), *n* = 26	NR	30.2 (9.8)	3.85%	IQ mean 83.5 (SD 18.2)	Specialist PICU	SCZ (DSM-IV-TR) 61.54%	NR
2010									
Hutton, 2008	UK	ASD (ICD-10), *n* = 135	NR	34.9 (21–57)	22.96%	IQ > 30 in all patients	Specialist outpatient	0%	NR
Mouridsen, 2008a	Denmark	Atypical autism (ICD-10), *n* = 89	NR	45.3 (7.2)	34.83%	IQ < 50, *n* = 20,	Specialist inpatient	34.80%	NR
						IQ > 50, *n* = 68			
Mouridsen, 2008b	Denmark	Infantile autism (ICD-9), *n* = 118	NR	5.4 (2.5), Range = 2–15	26%	IQ < 50, *n* = 48	Specialist inpatient	6.60%	NR
						IQ 50–69, *n* = 30 IQ > 69, *n* = 32			
Stahlberg, 2004	Sweden	ASD (ICD-10/DSM-IV), *n* = 129	GAF = 45.8 (10.8)	30.6 (9.7)	38.75%	IQ mean 86.2	Specialist outpatient	SCZ 2.94%;	NR
						(SD 21.3)		SCZ or other psychotic disorders (non-bipolar) 7.8%	
Volkmar, 1991	US	ASD (DSM-III-R), *n* = 163	NR	24.1 (5.58)	14.72%	NR	Specialist outpatient	SCZ (DSM-II-R) 0.61%	NR
				Range = 15–41					

**Legend: AD**: Autistic Disorder; **ASD**: Autism Spectrum Disorder; **CARS:** Childhood Autism Rating Scale; **GAF**: DSM-III-R Global Assessment of Functioning scale; **K-SADS-E**: Schedule for Affective Disorders and Schizophrenia for School-Age Children; **IQ**: Intellectual Quotient; **NR**: Not Reported; **PAC**: Psychopathology in Autism Checklist; **SCZ**: Schizophrenia. * The PAC encompasses five subscales: Psychosis: 10 items covering positive and negative symptoms and disorganisation. Depression: 7 items covering mood, cognitive, psychomotor and somatic symptoms. Anxiety disorder: 6 items covering psychological arousal, avoidance and cognitive symptoms. Obsessive-compulsive disorder, OCD: 7 items covering rituals, repetitive behaviour and obsessions. General adjustment problems (GAP): 12 items covering passivity, unrest, sleep problems, social avoidance, and aggression, self-harm and increased ritualising. Each item was given a score from 1 to 4 (1 = no problem; 2 = minor problem; 3 = moderate problem; 4 = severe problem).

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
