# Peer review of "Prevalence of Non-Affective Psychoses in Individuals with Autism Spectrum Disorders: A Systematic Review"

_jcm, 2019, doi:10.3390/jcm8091304_

Round 1

Reviewer 1 Report

Introduction

The introduction is well-written. The authors pointed out that the rate of non-affective psychosis in ASD highly varied across studies. This is a good reason for conducting a meta-analysis to quantitatively synthesize existing evidence, and that’s what the authors did. I wonder why the authors did not call this study a meta-analysis.

Methods

What were the keywords used to search for the articles? What statistical software was used to conduct the analysis? Random effects model or fixed effects model was used? What were the inclusion and exclusion criteria? Did the authors examine heterogeneity of the studies?

Results and discussion

PRISMA flowchart is mentioned but missing Table 1 is missing For the subgroup analyses, are the subgroups significantly different? How many studies included in this review used representative samples? This is important as it will influence your conclusion. If a mix of representative samples and non-representative samples were included, it would be helpful to conduct a subgroup analysis. Also need to address this in the limitation section.

Something minor: the in-text references in the manuscript should be superscript. I am not sure if this is the problem of word processing.

Author Response

Reviewer 1

- The introduction is well-written. The authors pointed out that the rate of non-affective psychosis in ASD highly varied across studies. This is a good reason for conducting a meta-analysis to quantitatively synthesize existing evidence, and that’s what the authors did. I wonder why the authors did not call this study a meta-analysis.

Authors’ response

Thank you, we decided to call it a pooled analysis rather than a meta-analysis as it is a pooled weighted prevalence; however, should the Editor prefer to call it meta-analysis, we would be happy to have the wording changed.

-Methods

What were the keywords used to search for the articles?

Authors’ response

Please, refer to file: Appendix A – Search strategy.

What statistical software was used to conduct the analysis?

Authors’ response

Thank you, added to page 3 line 99: “We calculated the weighted pooled prevalence mean for all studies using STATA 15 and we performed subgroup analyses by computing the weighted pooled prevalence… »

Random effects model or fixed effects model was used?

Authors’ response

Thank you, a random/fixed effect model would not be suitable for a pooled prevalence.

What were the inclusion and exclusion criteria?

Authors’ response

Thank you, added to page 2 line 84-87: “We included studies that evaluated the prevalence of non-affective psychoses in individuals with ASD with no age limit, but we excluded studies on mood disorders with psychotic features such as major depression with psychotic symptoms and bipolar disorder with psychotic symptoms”.

Did the authors examine heterogeneity of the studies?

Authors’ response

We agree with the reviewer that discussing heterogeneity is an important issue. We addressed heterogeneity by reporting major differences between study’s prevalence by narratively reviewing studies showing high prevalence versus low prevalence as reported in our limitations on page 7 line 288-290: “We observed significant heterogeneity among the 14 studies that were included, which is why we narratively discussed each study individually and performed subgroup analyses to achieve more homogeneous samples”.

-Results and discussion

PRISMA flowchart is mentioned but missing.

Authors’ response

Please, refer to file: Figure 1.

Table 1 is missing.

Authors’ response

Table 1 appeared cut in the manuscript file; please, refer to file: Table 1.

For the subgroup analyses, are the subgroups significantly different?

Authors’ response

Thank you, as there is no comparison group, it is not possible to estimate p-value (“significance”) for the subgroup analyses.

How many studies included in this review used representative samples? This is important as it will influence your conclusion. If a mix of representative samples and non-representative samples were included, it would be helpful to conduct a subgroup analysis. Also need to address this in the limitation section.

Authors’ response

Please, refer to page 6 line 248-249: “The subgroup analysis for “sample size” showed higher prevalence in smaller studies (N < 100: 19.6%, 95%CI= 3.4 to 35.9) against larger studies (N >/= 100: 6.36%, 95%CI= 0.0 to 13.2)” and to page 7 line 305-308: As previously discussed, it has been argued that sample sizes below 100 may not capture the real prevalence on non-affective psychoses in ASD19; we found a prevalence of 19.6% in smaller studies versus 6.3% in those with N >/= 100, in accordance with the existence of an overrepresentation bias in smaller studies”.

-Something minor: the in-text references in the manuscript should be superscript. I am not sure if this is the problem of word processing.

Authors’ response

Thank you, this does not depend on the file we submitted, but from the journal’s editing process.

Reviewer 2 Report

I think this is a good study and in line with previous results about the co-existence of non-affective psychoses and ASD, although this study found 2.5% less prevalence than previously reported (from 12% down to 9.5% or so).

Limitations in such studies are the age and other issue that some were addressed by the authors. 

Unfortunately several studies had to be excluded due to not fitting into the inclusion criteria of this study and os those several had to be excluded due to other reasons. Part of the difference in the prevalence might be due to this.

Nevertheless, the study again raises the importance of good clinical investigation and not missing the psychotic symptoms in ASD patients for better treatment of the patients which is valuable. 

Author Response

We thank the reviewer for the comments to our manuscript.